# Safety and data quality of EEG recorded simultaneously with multi-band fMRI

**Maximillian K. Egan**[1,2]☯*, **Ryan Larsen**[2]☯, **Jonathan Wirsich**[2,3], **Brad P. Sutton**[2,4], **Sepideh Sadaghiani**[1,2]

**1** Psychology Dept., Univ. of Illinois At Urbana-Champaign, Urbana, IL, United States of America,
**2** Beckman Institute for Advanced Science and Technology, Univ. of Illinois At Urbana-Champaign, Urbana, IL, United States of America, **3** EEG and Epilepsy Unit, Univ. Hospitals and Faculty of Medicine of Geneva, Geneva, Switzerland, **4** Bioengineering Dept., Univ. of Illinois At Urbana-Champaign, Urbana, IL, United States of America

☯ These authors contributed equally to this work.
\* mke3@illinois.edu

**Data Availability Statement:** All relevant data are available from the Illinois Data Bank at the following DOI: https://doi.org/10.13012/B2IDB-1484994_V1.

**Funding:** SS - NIMH R01MH116226 - National Institutes of Mental Health - https://www.nimh.nih.

## Abstract

### Purpose

Simultaneously recorded electroencephalography and functional magnetic resonance imaging (EEG-fMRI) is highly informative yet technically challenging. Until recently, there has been little information about EEG data quality and safety when used with newer multi-band (MB) fMRI sequences. Here, we measure the relative heating of a MB protocol compared with a standard single-band (SB) protocol considered to be safe. We also evaluated EEG quality recorded concurrently with the MB protocol on humans.

### Materials and methods

We compared radiofrequency (RF)-related heating at multiple electrodes and magnetic field magnitude, $B_{1+RMS}$, of a MB fMRI sequence with whole-brain coverage (TR = 440 ms, MB factor = 4) against a previously recommended, safe SB sequence using a phantom outfitted with a 64-channel EEG cap. Next, 9 human subjects underwent eyes-closed resting state EEG-fMRI using the MB sequence. Additionally, in three of the subjects resting state EEG was recorded also during the SB sequence and in an fMRI-free condition to directly compare EEG data quality across scanning conditions. EEG data quality was assessed by the ability to remove gradient and cardioballistic artifacts along with a clean spectrogram.

### Results

The heating induced by the MB sequence was lower than that of the SB sequence by a factor of 0.73 ± 0.38. This is consistent with an expected heating ratio of 0.64, calculated from the square of the ratio of $B_{1+RMS}$ values of the sequences. In the resting state EEG data, gradient and cardioballistic artifacts were successfully removed using traditional template subtraction. All subjects showed an individual alpha peak in the spectrogram with a posterior topography characteristic of eyes-closed EEG. The success of artifact rejection for the MB sequence was comparable to that in traditional SB sequences.

gov/ MKE - NSF Graduate Research Fellowship
Program in Cognitive Neuroscience - National
Science Foundation - https://www.nsfgrfp.org/
MKE - Beckman Institute Graduate Research
Fellows Program - Beckman Institute for Advanced
Science & Technology - https://beckman.illinois.
edu/ The funders had no role in study design, data
collection and analysis, decision to publish, or
preparation of the manuscript.

**Competing interests:** The authors have declared
that no competing interests exist.

## Conclusions

Our study shows that $B_{1+RMS}$ is a useful indication of the relative heating of fMRI protocols. This observation indicates that simultaneous EEG-fMRI recordings using this MB sequence can be safe in terms of RF-related heating, and that EEG data recorded using this sequence is of acceptable quality after traditional artifact removal techniques.

## Introduction

As the technologies available to modern cognitive neuroscience advances, the access and practicality of utilizing multiple imaging modalities simultaneously has increased dramatically. Complimentary pairings of imaging techniques allows researchers a far more comprehensive view of the brain by overcoming the limitations of the individual modalities [1]. Simultaneously recorded electroencephalography (EEG) and functional magnetic resonance imaging (fMRI) are two such complimentary techniques; EEG offers high temporal resolution in the millisecond range while fMRI provides high spatial resolution in the order of mm³, thus optimally capturing different types of neural activity [2]. Coupled with the non-invasive nature of recording and ease of access to equipment, the use of simultaneous EEG-fMRI is becoming more commonplace. The effectiveness of simultaneous EEG-fMRI has already been demonstrated for advancing the understanding of neuropsychiatric disorders [1], sleep [3–5], epilepsy [6,7], physiological rhythms [8], evoked activations [9–11], and ongoing brain activity and connectivity [12–14].

As a recent advance, concurrent EEG-fMRI imaging has begun to strongly benefit from simultaneous multi-band (MB) imaging in fMRI. In MB imaging multiple slices of the brain are acquired simultaneously. MB fMRI offers higher temporal resolution than the traditional SB sequences at little cost to fMRI signal-to-noise ratio (SNR) [15]. MB sequences have been previously shown to have comparable, if not greater functional SNR than non-MB sequences [16–18]; this is facilitated by the greater sampling rate, or lower TR, which allows more acquisitions per unit time.

However, there are many technical challenges in acquiring high quality EEG-fMRI data, and little work has assessed how these challenges are further affected by the use of MB fMRI sequences. The core challenge of concurrent recordings lies in the interaction between the EEG equipment and the fMRI scanner environment, causing artifacts in both modalities unique to simultaneous recordings. The artifacts in fMRI are minor, comprising a small decrease in cortical SNR due to increases in static magnetic field inhomogeneities near the electrodes, but without significant effects on sensitivity to signal changes [19–22]. However, the acquired EEG data shows strong gradient artifacts (GA) produced by the magnet's gradient switching, ballistocardiographic (BCG) pulse artifacts caused by small movements of the body/ electrodes due to cardiac pulsation combined with the Hall effect (production of a potential difference across an electrical conductor when a magnetic field is applied perpendicular to the flow of the current), and motion artifacts, especially in posterior electrodes [19,22]. A great deal of effort has been spent in the past two decades on the proper removal of these artifacts [22–27]. However, whether the methods previously developed to mitigate GA and BCG artifacts in concurrently recorded EEG are successful for newer MB fMRI sequences is insufficiently investigated.

Previous investigations into quality of EEG recorded concurrently with MB fMRI have demonstrated suitable data quality for several experimental conditions. Specifically, Foged

et al. [19] found no adverse effects on data quality using MB factors 4 and 8 (TR of 450 and 280 ms, respectively) and traditional GA and BCG artifact rejection methods. Uji et al. [28] used a silent recording paradigm, i.e. periods of active scanning interleaved with periods without active pulses (MB factor 3, TR = 3000 ms including 2250 ms silence). They showed that the EEG data acquired in the silent period was of high enough quality to investigate the EEG gamma frequency band that is otherwise particularly affected by the GA. Chen et al. [29] showed that compared to a SB sequence, a MB fMRI sequence (MB factor 4, TR = 550 ms) had minimal differences in EEG channel variance and spectra, and improved statistical and spatial sensitivity for resting state fMRI scans with a lower scanning duration. However, the free parameters used in MB sequences change with the needs of individual experiments. Therefore, to establish the general feasibility of obtaining acceptable EEG quality concurrently with MB fMRI it is critical to extend these few studies using other parameter sets optimized for other experimental needs. We chose our parameter set so as to scan the whole cerebrum at a short TR (MB factor 4, TR = 440 ms) without exceeding an MB factor of 4, while sparing the EEG alpha frequency band (~8–12 Hz) from residual RF excitation repetition artifacts (appearing at 15.9 Hz for our sequence, see methods).

More important than data quality when considering a new MB EEG-fMRI sequence is ensuring subject safety. For EEG-fMRI the key safety concern is the deposition of radiofrequency (RF) power that causes heating in the EEG leads and electrodes. While several studies have shown acceptable heating during simultaneous EEG-fMRI with both standard and high-field magnets using conventional SB scanning [20,30–32], only a few studies have demonstrated safe RF heating using MB EEG-fMRI [19,28,29,33], making research into MB EEG-fMRI safety a critical necessity.

Electrode heating has previously been characterized as a function of the Specific Absorption Rate (SAR) [19,32]. However, the whole-body SAR estimates can differ amongst scanners [34,35], rely on assumptions about the body being scanned [36], and typically provide biased measurements [34]. Given these problems it has been suggested to characterize safety limits using $B_{1+RMS}$, which is a fixed characteristic of the sequence and protocol and depends on the time-averaged RF amplitude transmitted by the scanner [35]. $B_{1+}$ refers to the magnitude of the (positively rotating component of the) oscillating magnetic field generated by the RF coil, and RMS refers to a root-mean-square, or quadratic mean of the $B_{1+}$ field calculated over time. This quantity is proportional to the oscillating electric fields responsible for RF heating. The major manufacturer of in-bore EEG amplifiers, Brain Products, has recently begun to use $B_{1+RMS}$ to specify safety limits [37]. Because $B_{1+RMS}$ is a relatively new standard, its use to assess the safety of simultaneous fMRI-EEG has been limited to the most recent literature on the safety of MB sequences [28].

The heating associated with a given experiment can be characterized in absolute or relative terms. Absolute heating experiments attempt to measure the temperature change due to scanning for a given experimental condition. The accuracy of these measurements is complicated by the presence of gradual temperature drifts in the bore [19]. Because these drifts can cause considerable variability, multiple measurements are required to accurately quantify the small changes of temperature that occur during scanning [29].

By contrast, a relative measurement can be used to compare the heating of two sequences or protocols under identical experimental conditions, such as scanners, samples, and temperature drift of the bore [29]. Because these conditions will similarly affect both sequences, the ratio of heating will be relatively insensitive to them. If one of the sequences has been established as safe under a wide variety of experimental conditions, then the relative measurement can be used to show that a new sequence is likely to produce similar or less heating under a similar variety of conditions. Relative temperature measurements have the potential to isolate

the scaling relationships that drive differences in heating between sequences. They have been used to show that for a wide variety of sequence types, the heating increase linearly with RF deposition [32].

Most previous EEG-fMRI safety studies have reported the absolute heating of MB sequences during scanning [19,28,29]. The few reports that have included both MB and SB sequences have not related these differences to differences in $B_{1+RMS}$ [19,29].

Here we measure the relative heating of a MB sequence compared to a SB sequence that has been established as safe under a wide variety of conditions [38]. We compensate for temperature drift by alternating between sequences multiple times. We compare the measured heating ratio with the heating ratio that is expected based on $B_{1+RMS}$ values associated with the sequences.

Our experiments are performed using a MB fMRI protocol chosen to exploit the ability of MB sequences to obtain images at high temporal resolution (low TR) without compromising image resolution or coverage [39–41]. After demonstrating the safety of the protocol using a relative temperature measurement, we demonstrate acceptable EEG data quality using data obtained from human subjects during eyes-closed resting state. We evaluate the efficacy of GA and BCG artifact rejection using traditional artifact rejection methods and characteristics of the EEG power spectrum to determine acceptable EEG data quality.

## Methods

We completed two different experiments in this study; 1) a phantom recording for assessing electrode heating and safety using the MB sequence in simultaneous EEG-fMRI, and after establishing safety, 2) eyes-closed resting state recordings in human subjects utilizing simultaneous EEG-fMRI with the same MB sequence to assess EEG data quality. All data are publicly available at the Illinois Data Bank [42].

### Human subjects

Nine subjects (four female) underwent simultaneous EEG-fMRI recordings. All subjects were right-handed, had no history of neurological disorders, were not taking any medication for psychiatric disorders or disease, had no history of alcohol/drug abuse. All subjects gave written informed consent according to procedures approved by the Institutional Review Board of the University of Illinois at Urbana-Champaign.

### fMRI

**MRI equipment.**   Data were acquired on a 3 T Prisma scanner with a 64-channel head coil (Siemens, Erlangen, Germany). TR markers from the scanner used for EEG-fMRI artifact removal were relayed through a specialized hardware box, the RTBox [43], connected directly to the scanner. Scanner clock synchronization was achieved using a BrainProducts SyncBox (Gilching, Germany) connected directly via a cable to the scanner's 10 MHz clock.

MRI sequences. Anatomical sequence

Anatomical information was obtained using a high-resolution 3D structural MPRAGE scan (0.9 mm isotropic, TR = 1900 ms, TI = 900 ms, TE = 2.32 ms, GRAPPA factor = 2).

Multiband functional sequence

Our MB protocol was chosen to minimize TR, while still maintaining the resolution and coverage typically used for non-MB sequences. The parameters of the MB protocol were: TR = 440 ms, TE = 25 ms, flip angle = 40˚, 28 slices, MB factor = 4, excitation pulse duration = 5300 us, GRAPPA acceleration factor = 2, slice thickness = 3.5 mm, fat saturation: on, bandwidth = 2056 Hz per pixel, in-plane matrix size = 128 x 128, FoV = 230x230mm. Our

laboratory has a particular interest in alpha band oscillations to address cognitive neuroscience questions [12,44,45]. Therefore, the MB factor and slice number were chosen to maintain maximal brain coverage at a short TR while moving the RF excitation repetition artifact outside of the traditional EEG alpha band range (~8–12 Hz). Seven RF pulses (28 slices/MB factor of 4 = 7 slice groups) every 440ms corresponds to an RF excitation repetition at 15.9 Hz, which is well above the alpha frequency range. Our flip angle was selected to remain below the Ernst angle, which for $T_1$ = 1331 $m$s in the grey matter [46] and TR = 440 ms, is expected to be 44˚. Of note, although not part of the research goal of the current investigation, we include a complimentary report of a task-based functional localizer using the MB sequence in a single human subject (S1 Fig). In line with extensive prior literature [16,17], we observed good functional sensitivity with this sequence.

Single band functional sequence

We compared the MB sequence against a sequence chosen to comply with the maximum intensity limits recommended by the manufacturer of the EEG equipment (Brain Products; [38]), which we name SB. The parameters of SB were TR: 2000 ms, TE: 30 ms, flip angle: 90, 25 slices, slice thickness 4 mm, fat saturation: on, bandwidth: 2470 Hz per pixel, resolution: 92 x 92, FoV: 230x230mm. For the MB protocol, $B_{1+RMS}$ was 0.8 µT; whereas $B_{1+RMS}$ of the SB protocol was 1.0 µT. Note that $B_{1+RMS}$ is a property of the sequence and does not depend on the properties of the object being scanned. Because RF energy deposition scales as the square of $B_1$ [47], the MB sequence is expected to exhibit lower RF energy deposition than the SB protocol by a factor of $(8/10)^2$ = 0.64. This estimate is in line with time-averaged RF power deposition values reported by the scanner during scanning of a standard 2 liter aqueous Siemens phantom designed to mimic typical body coil loading, where the RF power was 3.2 for the MB protocol and 5.0 for the SB, with the ratio being 3.2/5.0 = 0.64. Given that the heating scales with the RF power deposition, it is expected that heating during the MB protocol will be lower than that of the SB protocol by a factor of 0.64. It is important to note that the square of the ratio of $B_{1+RMS}$ is only predictive of relative heating for comparisons of the *same* sample. Samples that load the coil differently due to differences in volume or conductivity could require more RF power and exhibit greater heating. Safety limits based on $B_{1+RMS}$ are based on experience with a variety of experimental conditions, such as hardware and body types.

## Data acquisition

**EEG equipment and setup for phantom and human experiments.** EEG data was recorded with a 64-channel EEG cap in the standard 10-10-20 montage from BrainProducts that included 61 scalp electrodes and 3 drop-down electrodes (HEOG, IO, ECG) using the manufacturer's BrainVision Recorder software (BrainVision Recorder; [48]). The cap was connected to two MR-compatible BrainVision 32-channel EEG amplifiers within the scanner bore, and all impedances were kept <5 kOhms. For all scans using EEG-fMRI the amplifiers and battery pack were strapped down and weighted with sandbags on a stabilizing sled from BrainProducts to reduce vibration artifacts. The EEG recording hardware was directly connected to the SyncBox, which was also connected to the MR clock signal, producing 'Sync On' markers to verify synchronization. The RTBox was directly connected to the scanner and placed scanner pulse markers in the EEG file at the time of delivery of every RF pulse. The scalp electrodes had 10 kOhm built-in resistors (5 at amplifier + 5 at tip) and were recorded with a 0.5 µV resolution. The drop-down electrodes had 20 kOhm built-in resistors (5 at amplifier + 15 at tip) and were recorded with a 10 µV resolution. All electrodes had a low cutoff filter of 10s, high cutoff filter of 250 Hz, and a sampling rate of 5000 Hz. This combination of microvolt resolutions, sampling rate, and cutoffs gave us the highest possible recording

resolution while avoiding amplifier overloading. Preventing overloading of the amplifiers is critical for EEG-fMRI to ensure that the peaks of the artifact can be detected for GA/BCG artifact rejection. Additionally, during all human subject runs the scanner's helium pump was turned off to eliminate vibration artifacts at 42 Hz from the pump.

**Setup of heating experiments in phantom.** During the electrode heating tests a watermelon 'phantom' was fit with the 64-channel cap. The watermelon provides a conductive surface and permits fine-grained control over impedances at electrode contacts [29,49]. We abraded the watermelon with sandpaper prior to placing the cap on it and applying electrolytic gel to the electrodes, which allowed us to maintain impedances <5 kOhm, thereby protecting the amplifiers from high voltages induced by the scanner. Given the small size of the watermelon, the ECG electrode was routed underneath the watermelon once and placed in between the HEOG and IOG electrodes to ensure that no loop was created within the magnet (Fig 1). The monitored electrodes were exposed to the ambient air, without cushioning or other barriers to heat transfer. Temperature changes during scanning were measured with a Luxtron 812 two channel fluoroptic thermometer (LumaSense Technologies, Ballerup, Denmark). Fluoroptic probes were placed in the conductive paste between the electrodes and the watermelon surface.

**Acquisition of heating data in phantom.** We measured the relative heating of the MB protocol vs. the SB protocol in three separate phantom experiments. For Experiment A, the temperature probes were placed beneath the ECG electrode and TP7. For Experiment B, the temperature probes were placed beneath the ECG electrode and TP8. For Experiment C, the temperature probes were placed beneath the ECG electrode and FP2. All of our experiments included measurements of the ECG electrode because this electrode has the longest lead and is therefore expected to absorb more RF energy and exhibit greater heating [50]. The other probes were placed beneath the TP7, TP8, and FP2 because they are close to the edge of the

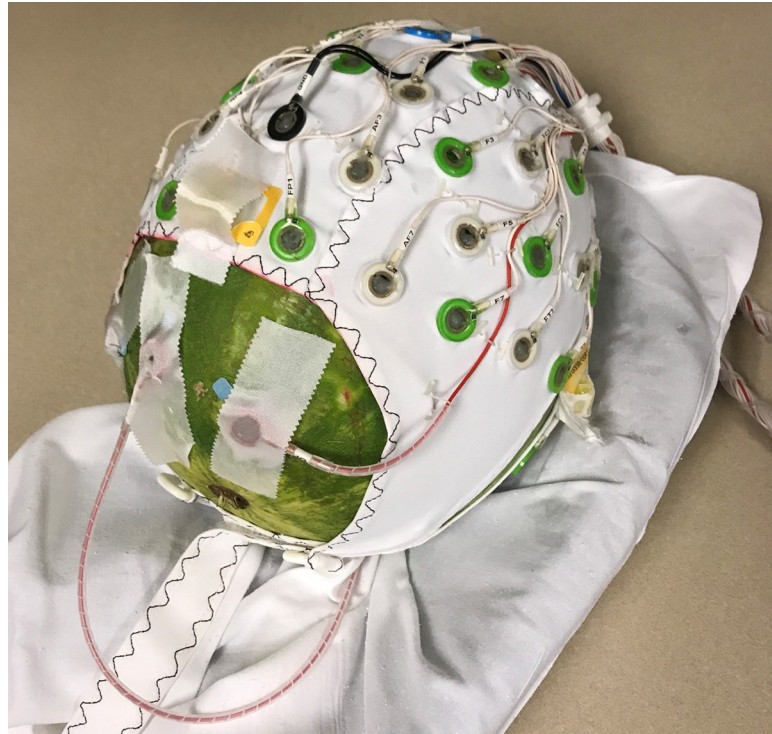

**Fig 1. Watermelon 'phantom' outfitted with the 64-channel BrainProducts EEG cap.**

cap and therefore easily accessible for placing the temperature probes, and they are spread across different areas of the head (electrodes correspond to left temporal, right temporal, and right frontoparietal, respectively). For each of the three heating experiments, the MB sequence was compared against the SB sequence by running each sequence three times, in an alternating fashion. The purpose of alternating the scans was to minimize bias due to long-term drift of the temperature. Each of the runs consisted of approximately 13.5 minutes of scanning and were spaced by approximately 5 minutes of rest between scans.

During the experiments, the fluoroptic thermometer performed two measurements per second; data were recorded by a computer connected to the Luxtron unit via a serial cable.

**Processing of phantom heating data.** Temperature data were processed using MATLAB® 2020a. Temperature measurements were smoothed using the "smoothdata" function using a Gaussian-weighted moving average filter with a window length of 200 seconds. The first pair of MB/SB runs was discarded from experiment 2 because scanning commenced before the phantom had equilibrated with room temperature. This was apparent from the fact that during the first run, and the rest periods before and after, the temperature of the probes decreased as the watermelon equilibrated with the colder environment of the scan room. Heating rates were calculated by dividing the difference in the smoothed temperature between the onset of the scan and the end of the scan by the scan duration.

**Acquisition of EEG data in humans.** We obtained EEG data with simultaneous fMRI from nine human subjects during eyes-closed resting state. More specifically, to assess the effect of MB imaging on EEG quality in extensive recordings, six of the subjects underwent two EEG-fMRI runs of 10 minutes on two separate days using the MB sequence (20 minutes total). To directly compare the quality of EEG acquired during the MB sequence, the SB sequence, and in the absence of active scanning sequences, we collected data from three additional subjects. The three subjects underwent 5 minutes of data acquisition inside the MRI scanner for each of the MB, fMRI-free, and SB conditions, in that order.

**Processing of human EEG data.** EEG data was preprocessed using the BrainVision Analyzer software (Version 2.2) [51]. Following standard procedures, GA subtraction was performed first followed by BCG artifact rejection as first done by Allen et. al [23,52]. The GA subtraction used marker detection from the trigger pulse markers obtained directly from the scanner (see methods) with a continuous artifact. A baseline correction over the whole artifact was used with a sliding average calculation of 21 marker intervals. Bad intervals were corrected with the average of all channels. The data was not downsampled (to minimize preprocessing). A lowpass finite impulse response (FIR) filter was applied at 100 Hz. The data was then segmented to include only artifact-free resting-state data, and BCG artifact rejection was performed using semi-automatic mode. After correcting all marked heartbeats, artifact removal of the heartbeat template was performed using sequential 21 pulse templates as the template average.

EEG data were analyzed in MATLAB 2018b® using EEGLAB (Version 2019.1) [53]. Prior to spectral analysis, the data was passed through a second lowpass FIR filter at 70 Hz. The spectrograms for scalp channels of each subject were computed at a frequency resolution of 0.2 Hz using the Multitaper approach. We chose not to do any further processing of the data (i.e., Independent Component Analysis decomposition) as we wanted to show the quality of the data with the minimum amount of cleaning resulting from the BrainVision Analyzer artifact rejections.

## Results

### Relative heating of the MB vs. SB sequences

To assure the safety of our MB sequence, we aimed at demonstrating that heating remained below that of the recommended SB sequence previously established to be safe [38]. In all

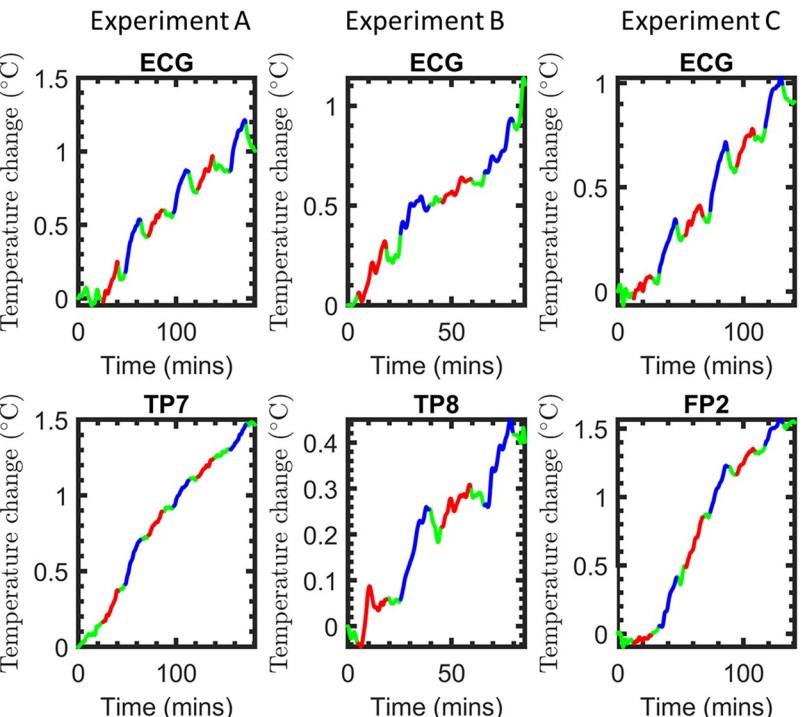

**Fig 2. Temperature measurements in the watermelon "phantom".** The total course of consecutive measurements during the different conditions are color coded for the MB sequence (red), the SB sequence (blue), and rest periods of no scanning (green). The ECG electrode, which has the longest lead and highest potential of heating, was included in all three experiments A through C. Experiments A, B, and C additionally measured temperature at TP7, TP8, and FP2, respectively.

experiments the temperature of the electrodes increased during scanning periods, as shown in Fig 2. Superposition of the temperature changes during each of the scans reveal consistently greater heating of the SB protocol for all electrodes, as shown in Fig 3. Average rates of heating were in the range 0.0068–0.021˚C/min, as shown in Table 1. The relative heating of the two sequences was estimated by dividing the rate of heating during each MB sequence with that of the SB sequence immediately following. Average values of these ratios from the four electrodes were in the range of 0.52 to 0.82, and the combined heating ratio from all electrodes is 0.73 ± 0.38, as shown in Table 1. The relative heating of the two sequences therefore is in approximate agreement with the RF power deposition ratio of 0.64 derived from the scanner (see Methods). This implies that the differences in heating between the two sequences are captured by the total RF power deposition.

Statistical analysis was performed by pairing all heating rates, $T'_{MB}$, of the MB sequences with the heating rates, $T'_{SB}$, of the SB sequence immediately following. Because pairs of heating measurements share identical conditions except the sequence, this analysis was performed by combining data from all electrodes. This pairing allowed for 8 pairs of measurements from each channel of the temperature probe for 16 total measurement pairs and 15 degrees of freedom. A paired $t$-test of MB sequences > SB sequences revealed significant differences, with $p$ = 0.0054, $t$-statistic = -3.25, confidence interval of $T'_{MB} - T'_{SB}$ = [-0.0093–0.0019], and estimated population standard deviation of 0.0069. Because the empirically measured heating was below that of the safe SB sequence, our MB sequence can be considered to be safe.

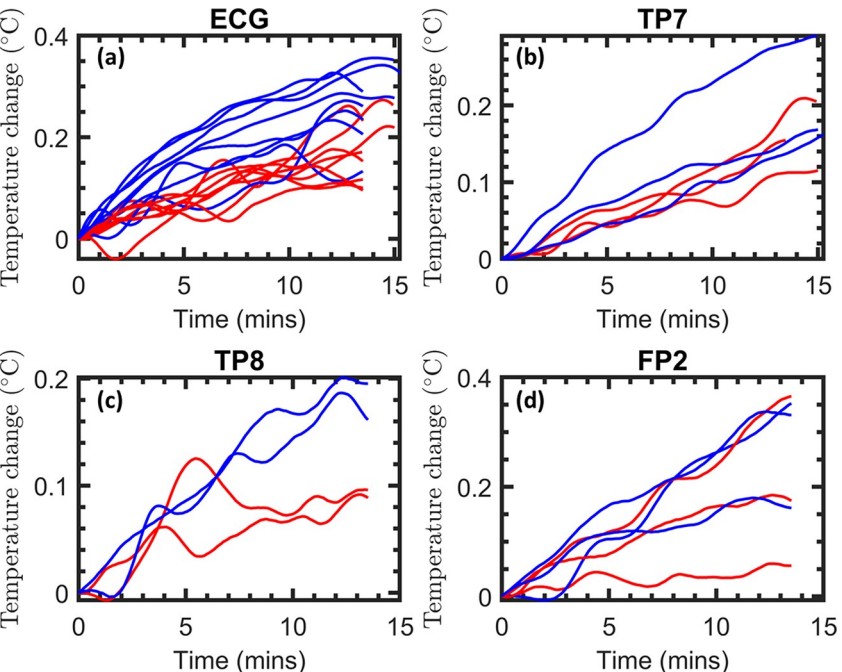

**Fig 3. Superposition of the individual temperature measurements in the watermelon "phantom".** Traces from the individual conditions that were recorded consecutively are superimposed for better comparison (MB sequence = red, SB sequence = blue).

## EEG data quality validation

To check EEG signal quality, we assessed the spectral dominance and topography of the alpha frequency band upon GA and BCG artifact cleaning. Alpha-band oscillations are uniquely positioned for this purpose because their exceptionally high power rises above the 1/f aperiodic component of the EEG spectrum during eyes-closed resting state. Fig 4 shows the log-power spectrogram of the EEG during eyes-closed resting state in a single subject (Subject 2). We demonstrate that each successive step substantially improves data quality, first showing the raw data without GA or BCG artifact subtraction (Fig 4A), then with only the GA artifact cleaned (Fig 4B), and finally the fully cleaned data with both GA and BCG artifacts removed (Fig 4C).The cleaned spectrogram showed a clear power peak in the alpha range (~10Hz) as expected during relaxed eyes-closed states, while displaying only a minimal residual power increase related to the GA at the RF excitation repetition frequency (Fig 4D).

Fig 5A shows the log-power spectrum for the 9 subjects averaged across posterior electrodes (O1, O2, Oz, PO3, PO4, PO7, PO8, POz) typically capturing particularly high power in the alpha range. For each subject we determined the individual alpha peak frequency at which power was highest in the ~8–12 Hz range [54] (denoted by a star for each subject in Fig 5A). At each subject's individual alpha peak frequency, a posterior topography (derived from all

**Table 1. Comparison of average heating of the MB and SB protocols at the ECG, TP7, TP8 and FP2 electrodes in the watermelon phantom.**

|  | ECG electrode (N = 8) | TP7 electrode (N = 3) | TP8 electrode (N = 2) | FP2 electrode (N = 3) | Combined (N = 16) |
|---|---|---|---|---|---|
| MB, Mean + SD of heating rates (˚C/min) | 0.012 ± 0.004 | 0.011 ± 0.014 | 0.0068 ± 0.0004 | 0.015 ± 0.01 | 0.012 ± 0.006 |
| SB, Mean + SD of heating rates (˚C/min) | 0.018 ± 0.004 | 0.014 ± 0.005 | 0.0132 ± 0.002 | 0.021 ± 0.008 | 0.017 ± 0.005 |
| Mean ± SD of MB/SB heating ratios | 0.72 ± 0.44 | 0.82 ± 0.18 | 0.52 ± 0.04 | 0.78± 0.54 | 0.73 ± 0.38 |

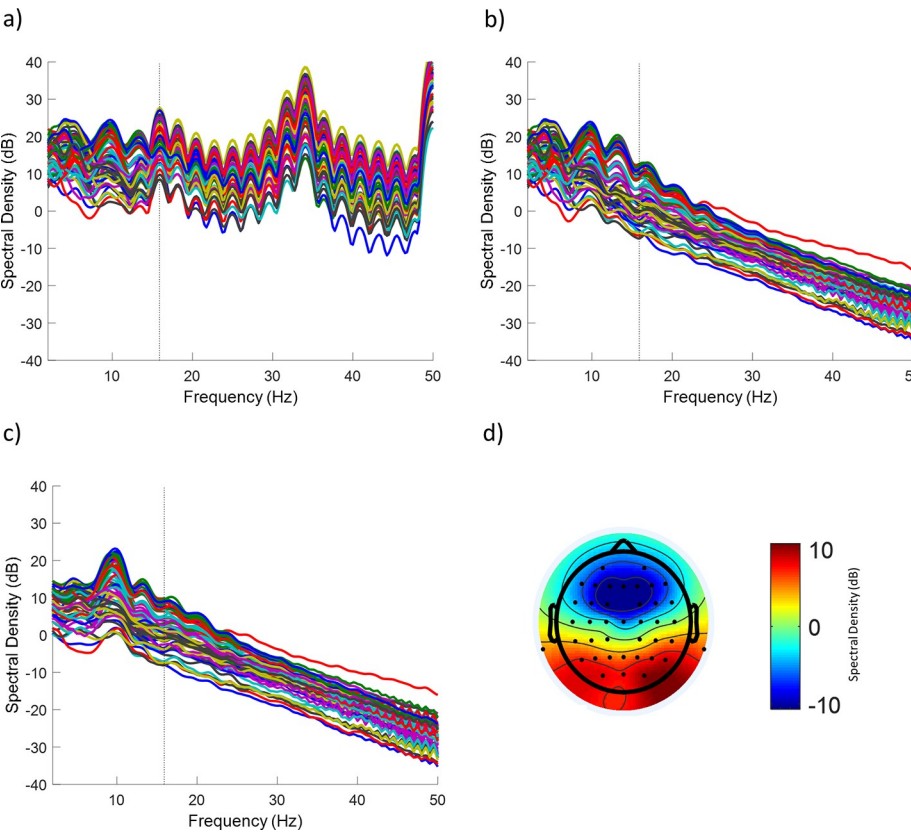

**Fig 4. Log-power spectrogram of the EEG during eyes-closed resting state in a single human subject during concurrent fMRI recordings with the MB sequence.** The spectrogram is shown for the (A) raw EEG data without artifact removal, (B) the EEG data with GA artifact rejected but prior to BCG correction, and (C) the cleaned EEG data after both GA and BCG artifact correction. Each trace corresponds to one of 60 scalp channels (1 excluded due to excessive noise). (D) shows the scalp topography at 10Hz for the cleaned EEG data. The prominent power peak at ~10Hz emerging more clearly after artifact correction (in C) and the posterior topography (in D) are consistent with the spectral dominance of the alpha rhythm in eyes-closed resting state. The dotted line represents the frequency of the RF repetition artifact at 15.9 Hz. This subject corresponds to Subject 2 in Fig 5.

channels) was detectable in all individual subjects (Fig 5B), while other power peaks in lower frequencies (particularly vulnerable to BCG artifacts) or at 15.9 Hz (the RF excitation repetition frequency) were greatly attenuated. The large spikes shown in Subjects 1 and 5 near 47 Hz were due to the scanner bore fan being on during the scan for subject comfort.

Fig 6 shows the direct comparison across the log-power spectrum of the MB, SB, and fMRI-free conditions for the final three subjects after removing GA and BCG artifacts. For all subjects the EEG recorded during the MB sequence showed a similarly clean power spectrum compared to the SB sequence. EEG in both fMRI conditions were comparable to the fMRI-free condition, except for an overall drop in power of matching magnitude for both sequences broadly across frequencies, a likely side effect of GA and BCG artifact subtraction. In particular, because the GA causes stronger contamination in the high EEG frequencies (>~20Hz, cf. Fig 4), the brain signal that can be recovered from this frequency range through GA removal is lower than in slower EEG frequencies [55]. Note that the loss of high-frequency EEG signals observed for the MB sequence were closely aligned with those observed in the SB sequence (Fig 6).

We conclude that EEG is of sufficient quality for cognitive neuroscience research using the MB sequence after application of artifact rejection methods originally developed for SB sequences, specifically template subtraction [23,52].

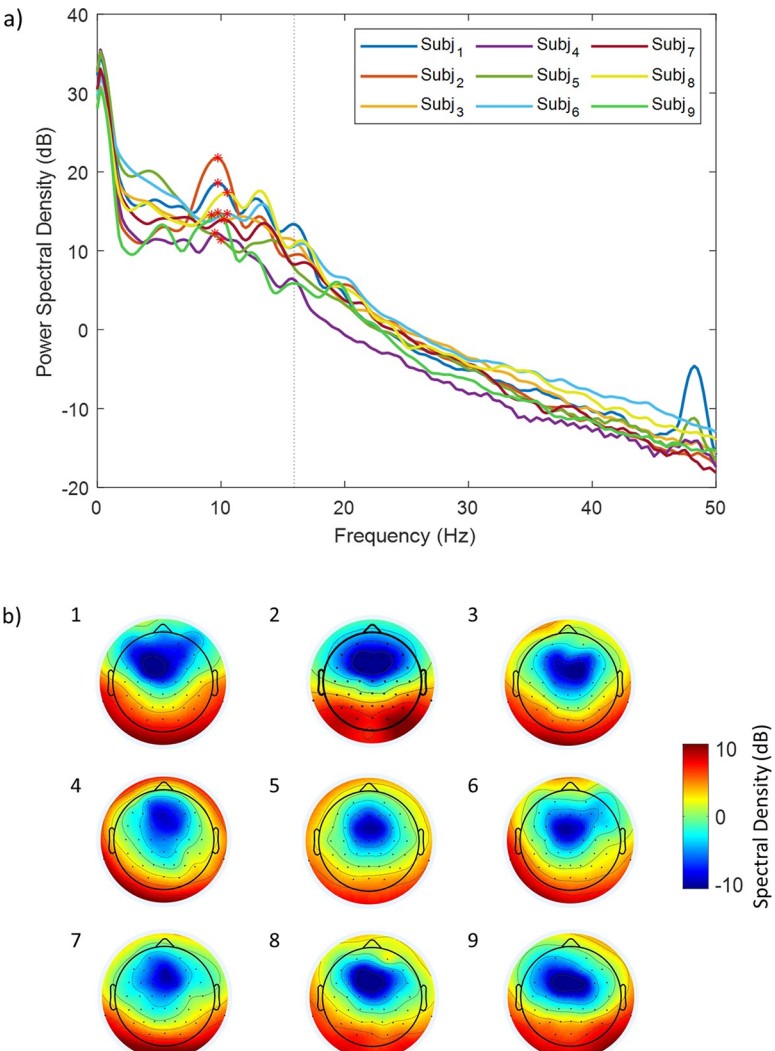

**Fig 5. EEG log-power spectrogram and topographies during eyes-closed resting state for all human subjects during concurrent fMRI recordings with the MB sequence.** (A) Power spectral density averaged across 8 posterior channels of each subject (O1, O2, Oz, PO3, PO4, PO7, PO8, POz). The stars denote the individual subjects' maximum values within the alpha band (8–12 Hz) i.e. individual alpha peak. The dotted line represents the scanner artifact at 15.9 Hz. (B) Corresponding scalp topographies using all channels for the 9 subjects at the individual alpha peak frequency.

## Discussion

Simultaneously recorded EEG-fMRI is a powerful tool that can provide information beyond what unimodal approaches are able to [56–58]. Although EEG-fMRI using traditional SB sequences is fairly well established, safety and data quality of EEG-fMRI imaging using modern MB fMRI sequences is less understood. Here we demonstrate that a particular MB sequence with high temporal resolution (TR = 440 ms) produces less RF heating at the EEG electrode sites than a traditional EPI sequence while maintaining acceptable EEG data quality.

Although previous studies have compared heating of a MB sequence with that of SB sequences [19,29], our study aimed to quantify the heating ratio of these sequences, and to show that for a given sample, this ratio depends on differences in $B_{1+RMS}$ between the protocols used. The rationale for this approach is that the use of safe $B_{1+RMS}$ limit of 1 µT has been

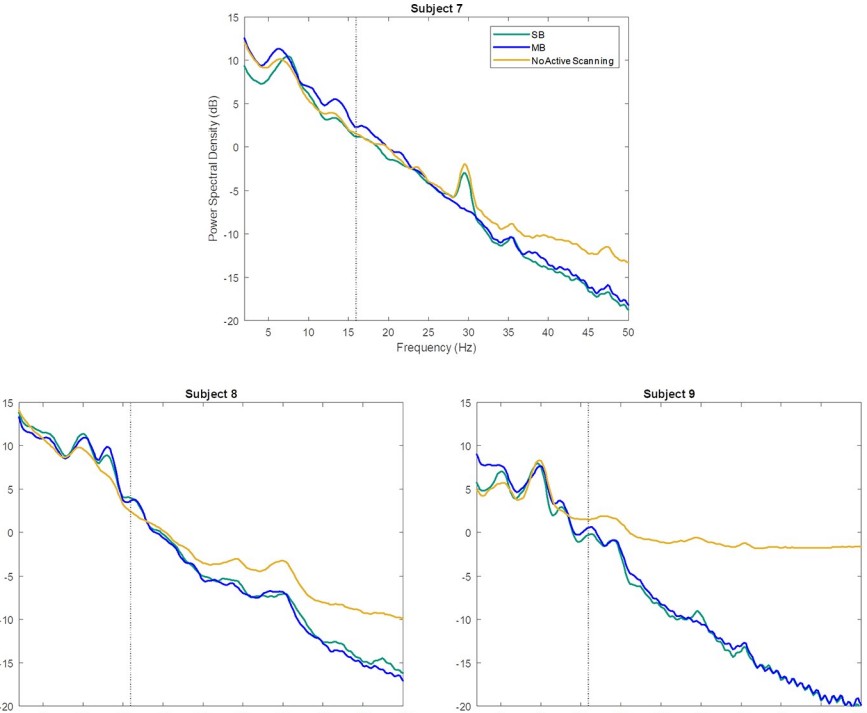

**Fig 6. Comparison of EEG log-power spectrogram across different concurrent fMRI conditions during eyes-closed resting state for three human subjects.** Power spectral density averaged across all channels directly comparing the MB sequence, SB sequence, and fMRI-free conditions for Subjects 7–9. The dotted black line denotes the RF excitation repetition frequency for the MB and SB sequences (15.9 Hz and 16 Hz respectively, indistinguishable on this figure).

developed over years of scanning of variety of subjects with different anatomical features and degrees of coil loading. Rather than attempting to reproduce this experience with a novel sequence, it is more practical to demonstrate that for a fixed sample the heating ratio is indeed governed by the square of the $B_{1+RMS}$ ratios. Such a result implies that the wide range of scanning conditions that have been shown to be safe with the SB sequences will also be safe for simultaneous EEG-fMRI acquired with MB sequences.

Based on scanner-reported values of $B_{1+RMS}$, we expected that, for a given sample, heating during our MB protocol would be lower than that of the SB protocol by a factor of 0.64. Using an interleaved strategy designed to minimize the effect of temperature drift, we empirically measured heating ratios in the range of 0.52–0.82 from the four electrodes. Standard deviations of the heating ratio ranged from 0.04 to 0.54. The low standard deviation of 0.04 for the TP8 electrode is likely an anomaly due to a sample size of two. The electrode for which we have the most data is the ECG electrode, with eight measurement pairs. For this electrode the mean and standard deviation of the heating ratio were 0.72 and 0.44. This is similar to the heating ratio measured from the combination of all electrodes, which is 0.73 ± 0.38. The theoretically expected heating ratio of 0.64 falls well within both ranges.

Variability of our measurements may be caused by local temperature fluctuations, perhaps arising from convective air currents, and measurement noise associated with limitations in the precision of the fluoroptic thermometer. Measurement noise limits our ability to draw conclusions about the spatial variation of the heating ratio. However, to the extent that heating ratio between sequences is determined only by the ratio of time-averaged RF power deposition, the shape of the spatial distribution of heating will be the same for both sequences, and the overall

magnitude of the heating distribution will increase linearly with the RF power deposition. This prediction is consistent with the expectation that the spatial distribution is largely determined by scanner hardware and anatomy [59].

In absolute terms, the heating rates we measured for all conditions were greater than 0.01 degrees C/min (see Table 1). These rates are higher than those observed in similar experiments [19]. These differences may be due to slow drifts in the ambient temperature, like those observed in some of our data during some of the rest periods (see Fig 2). The presence of drift is a serious complication for measuring the effect of scanning on absolute heating. Our results also demonstrate the potential of a relative measurement to compensate for drift, and the value of alternating between sequences multiple times while performing such measurements.

Our results support the usefulness of $B_{1+RMS}$ as a benchmark for assessing protocol safety. We have shown that values of $B_{1+RMS}$ can be maintained at acceptable levels even when using a MB factor of 4, a low TR of 440 ms, and 28 slices for whole-cerebrum coverage. This was achieved by using a relatively low flip angle of 40˚, and a moderately long pulse duration of 5300 µs.

Although temperature measurements were not performed on humans during scanning, our MB sequence has now been used with concurrent EEG in over 70 scan sessions at our local imaging center, without reports of heating sensations or burns from the subjects. Further, the three subjects that underwent simultaneous EEG-SB-fMRI and fMRI-free EEG in addition to EEG-MB-fMRI (cf. Fig 6 for EEG data) did not report any heating sensation in any of the conditions. These observations further support the safety demonstrated by the phantom heating experiment.

In addition to verifying heating safety, in our second experiment we conducted preliminary investigations into the success of GA and BCG artifact rejection for the EEG data acquired concurrently with our MB sequence. To this end, we assessed the spectral dominance and topography of the alpha frequency band due to its uniquely high and easily identifiable power during eyes-closed resting state. Indeed, after GA and BCG artifact rejection a clear posterior topography at a readily identifiable individual alpha peak frequency was observed in all six human subjects. Of note, due to our research interest in alpha oscillations [45] we chose the RF excitation frequency so as to spare this band from potential residual GA artifacts (see Methods). Particularly important, we saw the power at the RF excitation repetition frequency greatly attenuated after artifact rejection, implying that noise generated by the MRI sequence is not dominating the signal. Our observations in the EEG power spectrum indicate that traditional MR artifact rejection techniques [23,52] are sufficient for use in extended MB EEG-fMRI recordings. Additionally, we qualitatively demonstrate that there is no noticeable difference in the EEG power spectrum, e.g. at the respective RF excitation repetition frequency, between the MB sequence and SB sequence after artifact rejection, and that the spectrum is comparable post-artifact rejection to data obtained without active scanning. Future work would define EEG signal properties for quantitative comparisons of imaging sequences in a large sample.

To conclude, our results confirm the usefulness of in characterizing the relative heating of fMRI sequences. This work adds additional support to the growing body of literature on the safety and efficacy of multiband simultaneous EEG-fMRI imaging.

## Supporting information

**S1 Fig. Comparison of functional activations in a localizer task between the MB sequence and the SB-COV sequence.** The visualized contrasts correspond to sounds>implicit baseline (A), faces>objects (B), and objects>faces (C) (threshold for visualization purposes: voxel-wise p<0.001 uncorrected). For each contrast, top and bottom rows (MB and SB-COV sequence,

respectively) show equivalent slices for comparison. Activations comparable or stronger for the MB sequence are observed at the auditory cortex (A), the fusiform face area (B), and lateral occipital object-specific areas (C).
(PDF)

## Author Contributions

**Conceptualization:** Maximillian K. Egan, Ryan Larsen, Jonathan Wirsich, Brad P. Sutton, Sepideh Sadaghiani.

**Data curation:** Maximillian K. Egan, Ryan Larsen.

**Formal analysis:** Maximillian K. Egan, Ryan Larsen.

**Funding acquisition:** Sepideh Sadaghiani.

**Investigation:** Maximillian K. Egan, Ryan Larsen, Jonathan Wirsich.

**Methodology:** Maximillian K. Egan, Ryan Larsen, Sepideh Sadaghiani.

**Project administration:** Maximillian K. Egan, Sepideh Sadaghiani.

**Resources:** Jonathan Wirsich, Sepideh Sadaghiani.

**Supervision:** Sepideh Sadaghiani.

**Visualization:** Maximillian K. Egan, Ryan Larsen.

**Writing – original draft:** Maximillian K. Egan, Ryan Larsen, Sepideh Sadaghiani.

**Writing – review & editing:** Maximillian K. Egan, Ryan Larsen, Jonathan Wirsich, Brad P. Sutton, Sepideh Sadaghiani.

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
