## [Decision Letter · Decision Letter 0]

2 Dec 2020

PONE-D-20-25415

Safety and data quality of simultaneous EEG-fMRI using multi-band fMRI imaging

PLOS ONE

Dear Dr. Egan,

Thank you for submitting your manuscript to PLOS ONE. After careful consideration, we feel that it has merit but does not fully meet PLOS ONE’s publication criteria as it currently stands. Therefore, we invite you to submit a revised version of the manuscript that addresses the points raised during the review process.

We look forward to receiving your revised manuscript.

Kind regards,

Xi Chen

Academic Editor

PLOS ONE

Journal Requirements:

Reviewers' comments:

Reviewer's Responses to Questions

**Comments to the Author**

1. Is the manuscript technically sound, and do the data support the conclusions?

Reviewer #1: No

Reviewer #2: Yes

Reviewer #3: Partly

2. Has the statistical analysis been performed appropriately and rigorously? 

Reviewer #1: N/A

Reviewer #2: Yes

Reviewer #3: No

3. Have the authors made all data underlying the findings in their manuscript fully available?

Reviewer #1: No

Reviewer #2: Yes

Reviewer #3: Yes

4. Is the manuscript presented in an intelligible fashion and written in standard English?

Reviewer #1: Yes

Reviewer #2: Yes

Reviewer #3: Yes

5. Review Comments to the Author

Reviewer #1: The study investigates the temperature increase at an EEG electrode during simultaneous EEG and fMRI studies. It gives rise to several comments.

Major

The main concern is the steady and rather marked temperature increase during rest (no scanning) periods seen in figure 4, which remains unexplained. It would indeed be relevant to have the setup with temperature measurement performed without scanning a subject. However, results in the phantom (watermelon) are not reported.

The multi-band technique has lower SAR than the regular EPI technique, which is an unusual design choice and doubles B0-related slice distortions and chemical shift displacement artifacts. This is not typical for multi-band methodology. As such, studying the RF safety of the multi-band technique is inconsequential.

The description compensating the increased SAR at short TR by reducing the flip angle is misleading. The flip angle needs to be reduced to minimize signal saturation at short TR, which is customary for fast fMRI methods. Please clarify the Ernst angle at 3 T and TR: 440 ms.

Neither slice thickness nor readout bandwidth nor pixel size of the two fMRI methods are matched.

A single subject study to compare the sensitivity of two non-matched fMRI methods is not acceptable. Please clarify the reasons for considerably higher sensitivity in the case of MB-EPI. Was a correction for autocorrelations applied?

Please clarify how B1+ was measured and provide B1+ maps.

Reporting temperature for a single scalp electrode is inadequate. At least 4 temperature measurements should be performed at different locations of the scalp that are exposed to high B1+ levels.

There is no comparison of EEG data quality between EPI and MB-EPI.

In summary, this study does not provide sufficiently novel aspects that would advance the field of EEG-fMRI.

Minor

B1+ is not defined.

TP7 is a temporal lobe electrode, but this is not stated. The paper should be readable for a non EEG expert. Monitoring temperature at the ECG electrode is useful. However, placement and possibilities for heat exchange should be described.

A more detailed explanation is needed regarding the steady temperature rise seen in the rest (no scanning) periods in Fig.4. Is the room temperature changing? What are the possibilities for air exchange around the electrodes – is there a free space?

Fig 3 and 4: The blue and black lines are to close and difficult to distinguish.

Please clarify what MB-EPI sequence was used.

“Resolution” is the wrong description for the image matrix size.

Please clarify why an acquisition pixel size much smaller than the smoothing kernel width was used for this study.

Reviewer #2: The research is well planned and executed. The paper is well written, and the results are clearly presented. I have three things I wish the authors would elaborate on

1. I agree that ECG electrode is the one you definitively should measure in the heating experiment. However, in my opinion the reference 55 does not support the statement on line 271-272

“One of the probes was placed beneath the ECG electrode, because this electrode has the longest leads and is therefore expected to exhibit greater heating [55].”

Is the reference correct? The article by Angelone does not mention anything about the lead length and its impact on heating. Could you point it out, just in case I missed it. Does other research or guidelines of EEG equipment manufactured support the choice of measuring the heating of TP7?

2. Is the number of measured electrodes sufficient or not in the heating experiment? Are there enough repetitions? Can the results from two electrodes be generalize to all the electrodes? Mainly hoping for some discussion about the limitations of the study.

3. I wish that there would have been some statistical analysis of the heating rate and ratio (standard deviation). Or is there a reason for not doing it? Is there a significant difference between the measured and the theoretical value?

Minor things:

There are discrepancies in the text

-In the use of hyphen. In the beginning is used e.g. “MB-fMRI”, and in the abstract and latter part of the paper is used “MB fMRI”. I do not know which is grammatically correct.

- use of space before the physical quantity is sometimes missing e.g. 3000ms vs. 3000 ms. As far as I know, there should be a space between the number and the quantity.

-Decide which format to use in the text there are TR=3000, TR: 1900 ms. This even. varies within a sentence e.g. “0.9 mm isotropic, TR: 1900 ms, TI: 900 ms, TE = 2.32 ms, GRAPPA factor = 2) (line 155-156).

Line 51 SNR (declared later in the text on line 59)

Line 91 Maybe the abbreviation SAR should be within parenthesis (SAR).

Line 73: Maybe a typo in the number “including 2,250ms silence.”? Elsewhere numbers >= 1000 have been written without a comma.

Line 181 MNI space - reference missing?

Few abbreviations that have not been declared or assumed as common knowledge. Line 256 FIR and on line 263 ICA

Reviewer #3: The authors present a study where they first run a pilot MB protocol with a functional localizer task to evaluate MB-fMRI data quality (Figure 1). They then proceed to demonstrate the safety of simultaneous EEG/MB-fMRI with a capped watermelon with thermometers attached (Figures 2-4). Following from this, 6 human participants were recruited to be scanned for a total of 20 minutes resting-state eyes closed each (Figures 5-6). It is not immediately clear which “larger experiment” (refer to line 211) the authors are citing. There are several concerns that should be addressed:

Major Corrections

1. The authors claim that the MB sequence can be considered safe because the measured heating was below that of the “safe” SB-RF sequence (lines 296/297). As shown in Figure 4 (TP7), there was a continuous temperature rise – even during the “black line” periods (of no scanning). Were there any room temperature measurements – to potentially account for the continuous temperature rise?

2. The authors claim that the EEG data recorded is of acceptable quality (Line 35). However, the authors do not compare the EEG spectra to any ‘gold standard’ paradigm – as the authors rightfully say in lines 410-411, they could compare the EEG data quality between MB, SB, and scanner-off recordings. To properly justify their claim of acceptable EEG data quality, some more rigorous comparisons to an appropriate control would be needed.

3. Regarding thermal safety whilst scanning humans; it is somewhat surprising that thermal measurements were not conducted simultaneously. If they were conducted, they should definitely be presented to further illustrate the practical safety of simultaneous EEG/MB-fMRI – as participant effects (e.g. physiological mechanisms to maintain temperature) are not reproducible on a watermelon. If they were not conducted, perhaps the authors could also include whether any individuals felt any heating sensations or if any heating related incidents occurred – to further give reassurance of the safety of simultaneous EEG/MB-fMRI.

4. The present study does not really assess the data quality of the multiband fMRI data (whilst simultaneously recording EEG) – it only assesses the EEG data quality. Figure 1 reflects that of a pilot person of n=1 comparing between multiband and single fMRI scans – which appears fine. However, the authors could potentially go further to compare between the multiband fMRI and single band fMRI resting state scans they conducted with their n=6 main study – to see if there were any differences due to the different scan parameters. (For example, characterisation of resting state networks). This would justify the implication of the manuscript title – that the data quality of fMRI was also evaluated (during simultaneous acquisition of EEG/fMRI).

Minor Editorial Comments

- In Figure 3 and 4, consider capitalising the x-axis title “Time”, and including the “degrees” symbol in the y-axis title for ˚C

- In Table 1, perhaps the table caption could also include the word “average” – to indicate that it is the comparison of “average” heating

- The phrase including the word “sandbagged” in Line 220, whilst understandable in this context, could nonetheless be rephrased to avoid any malicious connotation. Consider changing to: “…were strapped down with sandbags to a stabilizing…”

6. PLOS authors have the option to publish the peer review history of their article (what does this mean?). If published, this will include your full peer review and any attached files.

Reviewer #1: No

Reviewer #2: No

Reviewer #3: No

---

## [Author Response · Author response to Decision Letter 0]

26 Feb 2021

Detailed responses to all reviewer comments can be found in the uploaded 'Response to Reviewers' file, filename 'EEG_fMRI_Heating_Safety_Resubmission_RevComments.docx'.

---

## [Decision Letter · Decision Letter 1]

30 Mar 2021

PONE-D-20-25415R1

Safety and data quality of EEG recorded simultaneously with multi-band fMRI

PLOS ONE

Dear Dr. Egan,

Thank you for submitting your manuscript to PLOS ONE. After careful consideration, we feel that it has merit but does not fully meet PLOS ONE’s publication criteria as it currently stands. Therefore, we invite you to submit a revised version of the manuscript that addresses the points raised during the review process.

We look forward to receiving your revised manuscript.

Kind regards,

Xi Chen

Academic Editor

PLOS ONE

Journal Requirements:

Reviewers' comments:

Reviewer's Responses to Questions

**Comments to the Author**

1. If the authors have adequately addressed your comments raised in a previous round of review and you feel that this manuscript is now acceptable for publication, you may indicate that here to bypass the “Comments to the Author” section, enter your conflict of interest statement in the “Confidential to Editor” section, and submit your "Accept" recommendation.

Reviewer #1: (No Response)

Reviewer #2: All comments have been addressed

Reviewer #3: All comments have been addressed

2. Is the manuscript technically sound, and do the data support the conclusions?

Reviewer #1: No

Reviewer #2: Yes

Reviewer #3: Yes

3. Has the statistical analysis been performed appropriately and rigorously? 

Reviewer #1: N/A

Reviewer #2: Yes

Reviewer #3: Yes

4. Have the authors made all data underlying the findings in their manuscript fully available?

Reviewer #1: Yes

Reviewer #2: Yes

Reviewer #3: Yes

5. Is the manuscript presented in an intelligible fashion and written in standard English?

Reviewer #1: Yes

Reviewer #2: Yes

Reviewer #3: Yes

6. Review Comments to the Author

Reviewer #1: The authors have addressed out questions and criticism. However, the study still gives rise to concern.

Major concerns

- Temperature measurements in a watermelon are not representative of in vivo conditions. The lack of in vivo temperature measurements greatly diminishes the impact of the study.

- The temperature drifts with and without EPI and MB-EPI make interpretation of safety related temperature increases very difficult. In a phantom these drifts should be controllable.

- The authors use a downgraded MB-EPI sequence with possibly distorted slice profiles and increased chemical shift displacement in the slice direction They do not characterize it in phantoms and in vivo: "We would like to emphasize that the single-subject investigation into BOLD sensitivity was only a confirmatory and supplementary report.”.

- What’s the point if the authors already know that MB-EPI SAR will be lower?: "Matching duration of the standard EPI and the multiband approach would result in higher SAR for the multiband approach, which would have counter-acted the point of the paper, to keep our multiband SAR below that of the standard, single band approach.”

There are problems with the meaning of the MR physics: “This length of RF pulse will not significantly degrade the slice profile, or create distortions that are worse than those that are routinely observed in the phase encoding direction….. B1+rms is a property of the sequence and does not depend on the properties of the object being scanned”.

The statement by the authors: "We agree with the reviewer that bandwidth and pixel size were different between scans…...However, the ... spatial resolution of this sequence is poor by contemporary standards" does not clarify and adjust the question regarding pixel size and readout bandwidth.

EEG power spectral density above about 20 Hz decreases when scanning. I am not sure whether that explains it: "a likely side effect of GA and BCG artifact subtraction”.

Reviewer #2: The authors have addressed all comments. In results section is presented some statistical analysis (paired t-test), but this is not presented in the Discussion part. However, I don't think this is an issue.

Reviewer #3: The authors have since included more phantom controls along with human subject controls, which much more appropriately justify the authors’ conclusions. This reviewer is largely satisfied with the edits; but have included several minor suggestions.

Suggestions

- Line 250: “…this electrode has the longest leads and is therefore…” Consider changing to: "...this electrode has the longest lead and is therefore..."

- Line 265-266: “The first pair of MB/SB runs was discarded from experiment 2 due to insufficient equilibration of the temperature probe with ambient temperature at the beginning of the scanning session.” How was “sufficient equilibration” determined? Perhaps a sentence to describe what exactly happened which led to the omission of data would be appropriate.

- Line 277: “fMRI-free” – please detail: was the EEG conducted on the person inside or outside the MRI scanner room? The reason why I ask is that this will determine whether the BCG artefact is still markedly present or not – and therefore improve replicability for future investigations.

- Line 323-325: The current problem with simultaneous EEG/MB-fMRI is that there is a heating concern when conducting simultaneous EEG/MB-fMRI – not when MB-fMRI is solely conducted without an EEG cap on the human participant. Have over 100 scan sessions with simultaneous EEG/MB-fMRI been performed already? (I assume not). I appreciate that the goal was to demonstrate the theoretical link between heating ratio and time-averaged RF power; but the practical safety of simultaneous EEG/MB-fMRI on human participants should still be assessed to absolutely reinforce the safety of simultaneous EEG/MB-fMRI. I assume that temperature readings were not conducted simultaneously with simultaneous EEG/MB-fMRI on the human participants (since they have not been included). However, instead of temperature readings, perhaps a comment could be made regarding the three human participants (as shown in Figure 6) who received simultaneous EEG/SB-fMRI, fMRI-free, and simultaneous EEG/MB-fMRI; and whether these three humans reported any differing heating sensations with the different scans. Whilst it is great that the safety is determined on watermelon phantoms, it is of pertinent importance that safety is determined on human participants.

7. PLOS authors have the option to publish the peer review history of their article (what does this mean?). If published, this will include your full peer review and any attached files.

Reviewer #1: No

Reviewer #2: No

Reviewer #3: No

---

## [Author Response · Author response to Decision Letter 1]

22 Apr 2021

Please see attached Response to Reviewers file ("EEGSafety_ReSub2_ReviewerComments.docx") for our detailed responses to the reviewer comments.

---

## [Editor Report · Decision Letter 2]

5 May 2021

Safety and data quality of EEG recorded simultaneously with multi-band fMRI

PONE-D-20-25415R2

Dear Dr. Egan,

We’re pleased to inform you that your manuscript has been judged scientifically suitable for publication and will be formally accepted for publication once it meets all outstanding technical requirements.

Kind regards,

Xi Chen

Academic Editor

PLOS ONE
---

## [Editor Report · Acceptance letter]

25 Jun 2021

PONE-D-20-25415R2 

Safety and data quality of EEG recorded simultaneously with multi-band fMRI 

Dear Dr. Egan:

I'm pleased to inform you that your manuscript has been deemed suitable for publication in PLOS ONE. Congratulations! Your manuscript is now with our production department. 

Kind regards, 

on behalf of

Dr. Xi Chen 

Academic Editor

PLOS ONE